# Comparison between Adult Patients with Sickle Cell Disease of Sub-Saharan African Origin Born in Metropolitan France and in Sub-Saharan Africa

**DOI:** 10.3390/jcm8122173

**Published:** 2019-12-09

**Authors:** Vasco Honsel, Djamal Khimoud, Brigitte Ranque, Lucile Offredo, Laure Joseph, Jacques Pouchot, Jean-Benoît Arlet

**Affiliations:** 1Internal Medicine Department, Sickle Cell Referral Center, Georges Pompidou European Hospital, AP-HP, 75015 Paris, France; vasco.honsel@aphp.fr (V.H.); djamal.khimoud@aphp.fr (D.K.); brigitte.ranque@aphp.fr (B.R.); jacques.pouchot@aphp.fr (J.P.); 2Paris Descartes Faculty of Medicine, Sorbonne Paris-Cité, 75006 Paris, France; 3Laboratory of Excellence GR-Ex, 75015 Paris, France; 4INSERM U970 PARCC Equipe 4 “Epidémiologie cardiovasculaire et mort subite”, 75015 Paris, France; lucile.offredo@inserm.fr; 5Department of Biotherapy, Necker Hospital, AP-HP, 75015 Paris, France; laure.joseph@aphp.fr

**Keywords:** sickle cell disease, Sub-Saharan Africa, acute chest syndrome, migration

## Abstract

Sickle cell disease (SCD) prevalence has increased rapidly in Europe as a result of an increase in the life expectancy of these patients and the arrival of SCD migrants from Africa. The aim of our study was to compare the phenotypes of adult patients born in Sub-Saharan Africa (SSA) who migrated to France with those of patients with the same origin who were born in France. This single-center observational study compared the demographic, clinical and biological characteristics of SCD adult patients of SSA origin who were born in France or SSA. Data were collected from computerized medical charts. Groups were compared using multivariate logistic regression with adjustment for age, gender and type of SCD. Of the 323 SCD patients followed in our center, 235 were enrolled, including 111 patients born in France and 124 patients born in SSA. SCD genotypes were balanced between groups. Patients born in Africa were older (median age 32.1 (24.4–39) vs. 25.6 (22.1–30.5) years, *p* < 0.001) and more often women (*n* = 75 (60.5%) vs. 48 (43.2%), *p* = 0.008). The median age at arrival in France was 18 years (13–23). The median height was lower among patients born in SSA (169 (163–175) vs. 174.5 cm (168–179), *p* < 0.001). Over their lifetimes, patients born in France had more acute chest syndromes (median number 2 (1–4) vs. 1 (0–3), *p* = 0.002), with the first episode occurring earlier (19 (11.6–22.3) vs. 24 (18.4–29.5) years, *p* < 0.007), and were admitted to intensive care units more often (53.3% vs. 34.9%, *p* = 0.006). This difference was more pronounced in the SS/Sβ0 population. Conversely, patients born in SSA had more skin ulcers (19.4% vs. 6.3%, *p* = 0.03). No significant differences were found in social and occupational insertion or other complications between the two groups. Patients born in SSA had a less severe disease phenotype regardless of their age than those born in France. This difference could be related to a survival bias occurring in Africa during childhood and migration to Europe that selected the least severe phenotypes.

## 1. Introduction

Sickle cell disease (SCD) is an autosomal recessive genetic disorder that occurs due to a mutation in the gene encoding the β-chain of hemoglobin (Hb), resulting in the production of an abnormal Hb called hemoglobin S (HbS). Polymerization of this pathological Hb causes chronic hemolysis that leads to chronic anemia and contributes to vasculopathy, followed by progressive development of chronic damage to the heart, kidney, brain, retina, bone and skin tissue. The disease is also characterized by the occurrence of painful bone vaso-occlusive crises (VOC) and acute chest syndrome (ACS), which are often triggered by cold, dehydration or infection, and increased susceptibility to infections. All of these complications lead to disability and early death [1]. Three predominant genotypes are responsible for SCD (S/S, S/C and S/β+/0 thalassemia). The most severe and frequent genotype (accounting for approximately 70% of patients in Sub-Saharan Africa (SSA) and France) is S/S (homozygous) SCD [1,2].

The mutation responsible for the disease appeared approximately 7300 years ago in the malarious regions, including SSA, the Arabian Peninsula and the Indian subcontinent [3]. From the 16th to 19th centuries, the slave trade exported the HbS allele to the Americas and the Caribbean. Since the 1960s, migrations of African populations with sickle cell trait to Europe, most frequently to former colonialist countries (France, United Kingdom and Belgium), has given rise to an increased incidence of newborns with SCD in these countries [4]. For the past 30 years, the arrival of African migrants with SCD has also increased the prevalence of patients with SCD in Europe.

In France (67 million inhabitants), approximately 400 newborns have been diagnosed with SCD each year by targeted neonatal screening since 1995, mainly in the Paris geographical area. More than 26,000 patients with SCD were estimated to be living in France in 2018 [5]. This prevalence has been increasing continuously at a rate of 5–7% per year since the 2000s in a proportion comparable to those of the United Kingdom (UK) and Germany [6,7]. Thus, France has become the country with the highest prevalence of SCD in Europe, followed by the UK [7]. Because of profound health inequalities, the countries with higher SCD prevalence rates have the worst prognoses. In SSA, where 75% of global SCD patients are born, more than 50% of homozygous (SS) children will die before the age of 5 years from sepsis (including malaria) or ACS according to some authors [8,9]. In contrast, in developed countries, the life expectancy of these patients has doubled over the last twenty years thanks to optimized care from birth [10,11,12].

In our adult SCD Referral Center located in Paris, France, 55% of our patients were born abroad, the majority of them in SSA. To the best of our knowledge, no study has compared the outcome of SCD patients born in an industrialized country to those born in SSA with the same genetic background as those who emigrated. Furthermore, the sociological characteristics of migrant SCD patients are widely unknown.

The main objective of this study was to compare the demographic, clinical and biological phenotypes of adult SCD patients born in Sub-Saharan regions to patients born in metropolitan France with parents of SSA origin. The secondary objective was to provide insights into a population of SCD migrant patients, who are under-represented in studies.

## 2. Patients and Methods

### 2.1. Study Design

We conducted a single-center, retrospective, observational study in the adult SCD Referral Center of the European Georges Pompidou University Hospital in Paris (France). For all SCD patients regularly followed in our center, socio-demographic characteristics, including the patient’s and his/her parents’ country of birth and educational level, clinical and laboratory data, and ongoing or past treatments, were retrospectively extracted from the electronic medical record charting system. These data were collected during routine follow-up outpatient visits using a standardized form and were regularly completed at each routine visit. Among the entire patient cohort, only SCD patients born in SSA or France with at least one parent born in SSA were included in the study. Thus, we excluded patients born in the French West Indies, French Guiana and Reunion because we assumed that their genetic backgrounds were different from that of the SSA population.

All participants were given written information explaining the aim of the study and the use of their medical file. They were given the possibility to express their refusal. The study protocol conformed to the ethical guidelines of the 1975 Declaration of Helsinki and was approved by the Comité d’Expertise pour les Recherches, les Études et les Évaluations dans le domaine de la Santé (CEREES) national commission under number 00001072. 

### 2.2. Specific Data Collection

Hematologic biological parameters were determined prior to hydroxyurea or transfusion program therapy at a steady state of the disease (when available). The last biological measurements of renal function and iron metabolism assessment, performed at a steady state of the disease, were recorded. For the renal function assessment, we used the Chronic Kidney Disease Epidemiology Collaboration (CKD-EPI) formula without adjustment for ethnicity [13]. Renal hyperfiltration was defined as an estimated glomerular filtration rate (eGFR) ≥130 mL/min/1.73 m^2^ for women and ≥140 mL/min/1.73 m^2^ for men [14]. Nephropathy was defined as chronic renal insufficiency or a urine albumin to creatinine ratio (ACR) >3 mg/mmol (at least 2 abnormal results over at least a 6 month period). For cardiac function assessment, systolic left ventricle (LV) dysfunction was defined by echocardiography as a LV ejection fraction <55% according to the definition used in adult SCD patients by Damy et al. [15]. A high tricuspid regurgitation velocity (TRV) was defined by a value ≥2.5 m/s, because higher values are predictive of cardiovascular mortality [15,16]. Cardiac involvement included systolic LV dysfunction, LV or left atrium dilatation measured by echocardiography, or clinical cardiac failure (occurrence of pulmonary edema). Cerebral vasculopathy was defined as the presence of cerebral vessel stenosis, Moyamoya disease, aneurysm or silent cerebral infarct and therefore was only evaluable in patients who had undergone brain magnetic resonance imaging (MRI) or a brain computed tomography (CT) scan. Thromboembolic events included phlebitis and pulmonary embolism. Transfusion-related adverse events were classified into 4 classes as follows: (1) detection of antibodies without a hemolytic reaction, (2) delayed hemolytic transfusion reaction (DHTR) with detection of antibodies, (3) DHTR without antibodies and (4) acute hemolytic transfusion reaction. Active or resolved HCV infections were defined as detection of anti-HCV antibodies with the presence or absence of HCV RNA in the serum. Active or resolved HBV infections were defined as detection of anti-HBc antibodies in the serum with or without HBs antigen. For patients who benefited from stem cell transplantation, the follow-up was censored at the time of the allograft. 

### 2.3. Statistical Analysis

Descriptive data are presented as percentages (95% confidence intervals (CIs)), medians (interquartile ranges) or means (standard deviations). The clinical, biological and demographic parameters of the two groups of patients were compared using univariate and multivariate logistic regression with adjustment for age, gender and type of SCD. We also performed the same comparisons in the SS/Sβ0 sub-population, with adjustment for age and gender. *p* values inferior to 0.05 were considered significant. All analyses were performed using R version 3.3.1 (R Core Team (2016) R: A Language and Environment for Statistical Computing. R Foundation for Statistical Computing, Vienna, Austria. (https://www.R-project.org/)). 

## 3. Results

### 3.1. Demographic Data

Two hundred and thirty-five SCD patients with SSA origin were included in the study (Figure 1), including 111 patients born in metropolitan France and 124 patients born in SSA. The country of origin of all patients and country of birth of the migrant patients are presented on Figure 2. Almost half of the patients born in SSA migrated from three African regions: Congo (Republic of the Congo and Democratic Republic of Congo), *n* = 32 (25.8%), Cameroon, *n* = 19 (15.3%), and Ivory Coast, *n* = 14 (11.3%). The parents of the patients born in France were from West Africa in 68% of cases and from Central and Eastern Africa in 29% of cases. Regarding the parents of patients born in SSA, the respective proportions were 62.2% and 37.7% (non-significant (NS)). The median age at arrival in France was 18 (13–23) years.

The demographic, clinical and biological characteristics of the patients and the comparison between both groups are presented in Table 1. The median age at the last follow-up was higher in the sickle cell patients born in SSA than in those born in France (32.1 (24.4–39) vs. 25.6 (22.1–30.5) years, *p* < 0.001), as was the percentage of women (60.5% vs. 43.2%, *p* = 0.008). Therefore, we adjusted the statistical analyses for age and gender. 

The genotype frequencies were quite similar in both groups. Nevertheless, we have also adjusted the multivariate logistic regression analysis for the genotype (Table 1 to 5, last column) and also performed the analyses in the SS/Sβ0 sub-population (79.2% of the cohort) (Appendix A). 

Patients born in SSA were diagnosed later with SCD (median age 3 years (1–9.5), vs. 0 (0–2), *p* = 0.0001) and benefited less from neo-natal screening (9.2% vs. 50.6%, *p* = 0.001). Additionally, the heights of these patients were significantly lower. 

Education levels and occupational rates were not significantly different between the two groups. For example, the proportion of patients with higher education (those who studied beyond high school) was 54.9% in the patients born in SSA and 57.5% in the patients born in France (NS). The occupational rate at the last follow-up was 59.3% among the patients born in SSA and 43.8% for the others (NS). Biological measurements, performed at a steady state of the disease, were similar between groups (Table 1 and Appendix A).

### 3.2. Acute Clinical Events and Chronic Complications

Data related to acute complications are shown in Table 2. Patients born in France experienced more ACS episodes (median number of episodes: 2 (1–4) vs. 1 (0–3), *p* = 0.002), and their first ACS episode occurred earlier (median age 19 (11.6–22.3) vs. 24 (18.4–29.5) years, *p* = 0.007). These patients were also admitted to intensive care units (ICU) more often (53.3% vs. 34.9%, *p* = 0.006). 

Focusing on homozygous (SS) and Sβ0 patients, 86% of those born in France had presented at least one episode of ACS over their lifetime, compared to 70.7% of the SS/Sβ0 patients born in SSA (*p* = 0.005, adjusted for age and sex), even if the homozygous patients born in SSA were 4.4 years older (median age 30 (23–36.8) vs. 25.6 (22.2–30.2) years, *p* < 0.0001). SS/Sβ0 patients born in France also experienced more ACS episodes in their lifetime (3 (1–4) vs. 1 (0–3), *p* = 0.002) (Appendix A). 

In contrast, the number of admissions for VOC during the previous year and the prevalence rates of stroke and priapism were similar in the two groups.

Chronic complications are presented in Table 3. Retinopathy was diagnosed earlier in the patients born in France (median age 21.2 (16.7–25.7) vs. 28.9 years (24.2–34.3), *p* = 0.05), but the frequency of the use of laser photocoagulation was identical. The patients born in Africa were more likely to have skin ulcers (19.4% vs. 6.3%, *p* = 0.03), with the first episode at an earlier age. The prevalence rates of nephropathy, hyperfiltration, cerebral vasculopathy, retinopathy, heart and bone involvement were similar between the two groups. The results of the comparisons regarding chronic complications in the two groups were similar when analyzing the SS/Sβ0 subpopulation only (Appendix A).

### 3.3. Therapeutic Management

Data related to treatments provided during follow-up and treatment tolerance are presented in Table 4. The lifetime exposure to hydroxyurea (HU) was the same between the two groups (52.4% in patients born in SSA vs. 63.1% in patients born in France, *p* = 0.2), but HU was introduced later in patients born in SSA (median age 22.3 (16.4–30) vs. 18.6 (14.4–23.5) years, *p* = 0.002). No difference was found in the cumulative HU dose. Among the SSA native patients, 46.3% had been transfused in Africa prior to their arrival in France. The immune-mediated complication rates related to blood transfusion were comparable between the two groups. Proven blood transfusion-related hemochromatosis (by MRI or liver biopsy) was similar between the groups but was diagnosed earlier in patients born in France (median age 16.9 (13.8–20.4) vs. 25.3 (18.3–31.2) years, *p* = 0.006). Lifetime exposure to a regular blood transfusion program was not different between the groups.

Three SS patients benefited from haplo-identical stem cell transplantation (at 19, 20 and 29 years), and two of them were cured. The remaining patient had graft failure but did not present any new VOC requiring hospitalization during the 4 years of post-transplantation follow-up. One additional SS patient benefited from a successful geno-identical transplantation at the age of 28 years. Three of these four patients were born in SSA. 

The proportions of patients who were up-to-date with the pneumococcal vaccine at the last follow-up (according to the French guidelines) were not significantly different between the groups (43.4% among SSA natives and 29.7% among metropolitan France natives, *p* = NS).

### 3.4. Maternal and Fetal Complications

Women were older in the group born in Africa (median age = 34.1 (28.1–39.7) vs. 26.1 years (23.8–30.6), *p* < 0.0002). The median age at first menses was comparable (14 (13–16) vs. 14 (13–16), *p* = 0.4). No difference between groups was observed concerning miscarriages or gestational complications. The number of children per woman (adjusted for age) was significantly higher in the SSA population (median number of children: 1 (0–2.5) vs. 0 (0–0), *p* = 0.02). 

### 3.5. Comorbidities

The detailed data are provided in Table 5. Patients born in SSA were more likely to have hepatitis B serological markers compatible with chronic or previous viral exposure with resolution of infection (21.3 vs. 1.2%, *p* = 0.004).

## 4. Discussion

In this study, we compared for the first time the phenotypes of SCD patients born in metropolitan France with those born in Africa who migrated to France. Surprisingly, patients born in France who received optimal care from birth, including earlier treatment with hydroxyurea, had a more severe clinical phenotype than patients born in Africa who received optimal medical care only after their arrival in France at a median age of 18 years. This is observed despite the fact that some patients with SCD born in Africa who migrated to France may remain undiagnosed, especially for the compound heterozygous type of SCD, because of a less severe phenotype. Patients born in France presented a more frequent history of ACS over their lifetimes (78.2% vs. 62.8%), with the first episode at a younger age (5 years earlier in median) and more intensive care admissions, even though they were younger than their counterparts born in Africa; however, no differences in the main hematological tests, including the total Hb and HbF levels, were observed between the groups. Those findings may be explained by a “double selection” of patients born in Africa. The first selection step was positive selection of less severe phenotypes during childhood in Africa, where more than 50% of SCD children are estimated to die under the age of 5 years, mostly from infectious diseases, including malaria and ACS [17,18]. The observed tendency towards a decrease in the risk of ACS with an increase in the age of arrival in France is in favor of this hypothesis. The second selection step involved their journey to France. Those with a better social status enjoyed safer travelling conditions, and those who were less disabled were more likely to withstand the long and difficult emigration route.

Similarly, retinopathy was diagnosed earlier in the patients born in France. Notably, retinopathy can remain silent for a long time if not screened, especially in Africa where medical equipment and trained ophthalmologists are lacking. Whether the earlier diagnosis of patients born in France was associated with a less severe visual prognosis was unclear because blindness was very rare in our cohort, and the laser coagulation prevalence was identical. A study conducted in a referral center in Mali (*n* = 1064 patients, mean age of 24.2 years) in which patients were screened for retinopathy starting at an age of 10 years estimated its presence to be 8.8% in SCD individuals over the age of 10 years regardless of their genotype, with a maximal prevalence of 12.4% among S/C patients [19]. The prevalence reported in our Africa-born patients (49.1%) was much higher, but their mean age was 9 years older (33.2 ± 10.7). Conversely, leg ulcers were three times more common in the patients born in Africa than in the patients born in France, with the first episode at an earlier age (4.7 years earlier, in median). According to some studies, skin ulcers occur more frequently in homozygous patients living in tropical zones [20]. We observed that in two-thirds of the cases (*n* = 8/13 SS/Sβ0 patients born in Africa with available data), skin complications developed when the patients were already settled in France. However, we do not know whether the occurrence of ulcers was contemporary to journeys back to Africa. In this case, the ulcers could be related to late treatment of minor wounds or insect bites. Most of these patients (*n* = 10/15 SS/Sβ0 patients born in Africa with available data) have experienced skin ulcers before the first introduction of hydroxyurea. In comparison, all six SS/Sβ0 patients with skin ulcers born in France had experienced their first ulcer after the start of hydroxyurea (at a mean age of 19.5 years). However, among the patients born respectively in France or in SSA who experienced skin ulcers after the introduction of hydroxyurea (at a mean age of respectively 19.5 and 15.9 years), the mean age at the first ulcer was similar (respectively 26.6 and 25.5 years).

The access to hydroxyurea is, to date, unaffordable in SSA, due to its cost. Interestingly, our study shows no difference in lifetime hydroxyurea exposure between the groups, suggesting that SSA subjects are willing to take hydroxyurea, when it is reimbursed by a care system.

The higher frequency of HBV serologic stigma in patients born in Africa is consistent with a recent French multicenter study [21]. Arlet et al. reported a prevalence of HBV positive serology of 42.5% (*n* = 65/153 had HBc antibodies) for HBV with two-thirds of the infected patients born in SSA.

Growth retardation is common in homozygous SCD patients and is attributed to hypercatabolism, endocrine dysfunction and nutritional deficiencies [22]. Chronic transfusion therapy given for 24 months is effective in normalizing growth charts in young SCD patients, as is hydroxyurea treatment in industrialized countries [23,24,25]. Thus, we can hypothesize that easier access to transfusions or hydroxyurea during childhood explains the differences observed in our study. However, this explanation was not satisfactory since the growth retardation observed in SCD children was shown to usually correct itself without any intervention after puberty compared to the growth of controls living in the same area in the USA in the 1980s before hydroxyurea therapy and in Africa [26,27]. A more nutritious diet in France could be an alternative explanation for this difference. Height differences could also be related to the patients’ origins. Indeed, the average height varies from one African region to another [28].

The education level was identical between the groups. However, a significant number of the patients born in SSA were mostly educated in France (age of arrival was ≤ 13 years for 25% of these patients). In addition, it is possible that the SSA patients who were most likely to realize an emigration project and permanent settlement in France were more likely to receive education in their country of origin, compared to other patients having not emigrated. Finally, the occupation rates were similar between the groups, but the level of professional qualification or the salary was not recorded in our study.

The main limitation of this study is bias in data collection due to its retrospective nature, particularly for the SCD patients born in Africa. Most migrants were unable to present with their prior medical chart when they arrived in France; therefore, collection of the medical history was obtained by interview only and was limited by memory and cultural representation of the disease. Another limitation of this work is that we did not estimate the influence of hospitalizations on the occurrence of ACS. Since any hospitalization promotes the occurrence of ACS, and assuming that in Africa the prohibitive cost of medical care reduces hospital admissions, it can be hypothesized that lower hospitalizations in Africa are a confounding factor favoring (as regards number of ACS) patients born in Africa. However, other factors promoting the occurrence of ACS to the detriment of patients born in Africa were also not taken into account, like exposition to malaria, heat and dehydration, or lack of access to transfusions or hydroxyurea.

Finally, we cannot exclude a bias due to a genetic admixture with other populations in our patients born in France, as six of these 111 patients had one of their parents of another origin than SSA (five Caribbean, one Greece). Nevertheless, it is unlikely that such an admixture plays a significant role in regards to its weak prevalence (5.4%).

## 5. Conclusions

Compared to patients born in France, SCD patients born in SSA who migrated to France unexpectedly had a less severe clinical phenotype in terms of ACS regardless of their age, probably due to selection and survival bias. In contrast, skin ulcers occurred more frequently in patients born in SSA.

## Figures and Tables

**Figure 1 jcm-08-02173-f001:**
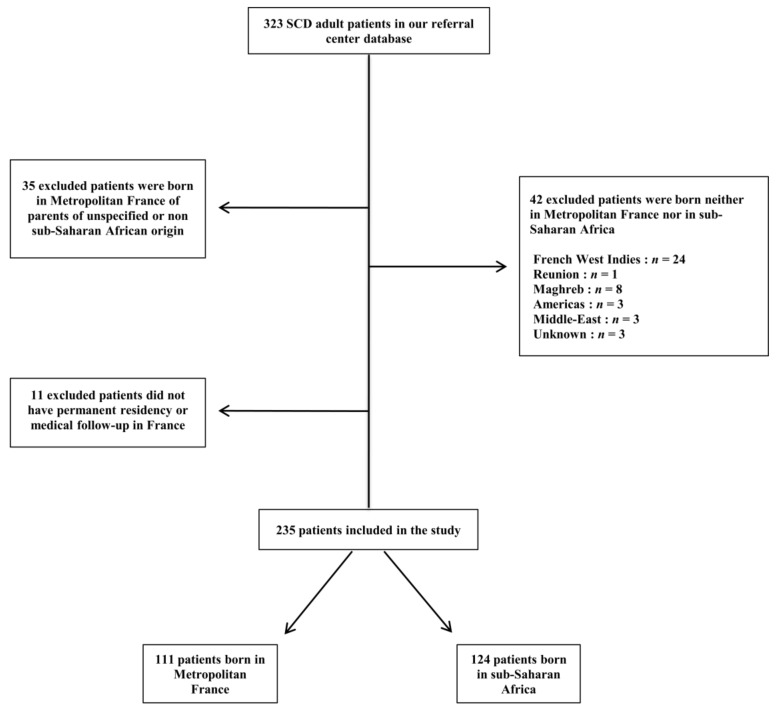
Flow-chart.

**Figure 2 jcm-08-02173-f002:**
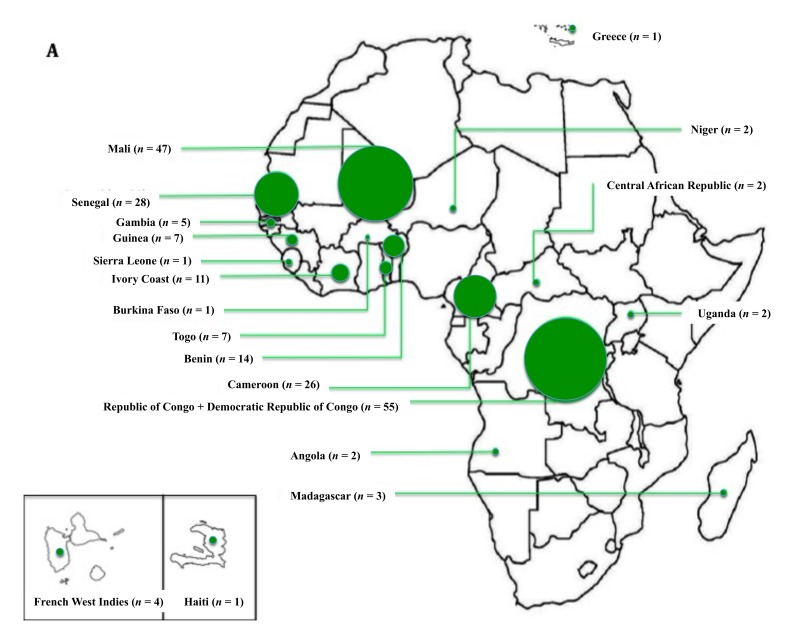
Countries of origin of all patients and countries of birth of the migrant patients. (**A**) Geographical origins of the parents of patients born in metropolitan France. (**B**) Geographical origins of the parents of patients born in Sub-Saharan Africa. (**C**) Countries of birth of the migrant patients. The circles are proportional to the number of individuals. *n* indicates the number of individuals for each country. If their parents were born in two different countries, one point was allocated to each of the two different countries. For better readability, the scale of the circles in (**C**) is twice that in (**A**) and (**B**). The geographical distribution of the parents’ countries of origin (**A** vs. **B**) was not statistically different (*p* = 0.084).

**Table 1 jcm-08-02173-t001:** Main characteristics of the sickle cell disease patients according to their birth region.

Demographic and Biological Parameters	Whole Population *n* = 235	Patients Born in France *n* = 111	Patients Born in Sub-Saharan Africa *n* = 124	*p* Value *	*p* Value **
**Male**, *n* (%)	112 (47.7)	63 (56.8)	49 (39.5)	**0.008**	0.02
**Age at last follow-up** (years)	29.2 (23–35.1)	25.6 (22.1–30.5)	32.1 (24.4–39)	**<0.001**	**<0.001**
**Age at SCD diagnosis** (years)	1 (0–5)	0 (0–2)	3.0 (1–9.5)	**<0.001**	**0.001**
**Neonatal screening**, *n* (%)	50/170 (29.4)	42/83 (50.6)	8/87 (9.2)	**<0.001**	**<0.001**
**Age at arrival in metropolitan France** (years)	-	-	18 (13–23)	-	-
**Duration of residency in France** (years)	-	-	12.6 (8.3–19.2)	-	-
**Height** (cm)	171.8 (165–178)	174.5 (168–179)	169 (163–175)	**0.001**	**0.001**
Male (cm)		176 (173.3–182)	176.6 (171.8–180)		
Female (cm)		169 (164.3–173.8)	165 (161–169)		
**Weight** (kg)	66.7 ± 12.4	67.6 ± 13.2	65.9 ± 11.8	0.02	**0.007**
Male (kg)		68.9 ± 11.6	68 ± 10.4		
Female (kg)		65.9 ± 14.9	64.5 ± 12.4		
**BMI** (kg/m^2^)	22.1 (20–24.6)	21.6 (19.9–24.2)	22.4 (20.3–24.9)	0.3	0.1
**Genotypes**					
S/S, *n* (%)	180 (76.6)	92 (82.9)	88 (71)	0.6	-
S/C, *n* (%)	40 (17)	16 (14.4)	24 (19.4)		
S/β^+^, *n* (%)	9 (3.8)	2 (1.8)	7 (5.6)		
S/β°, *n* (%)	6 (2.6)	1 (0.9)	5 (4)		
**Hb** (g/dL)	9 (8–10.7)	8.6 (7.6–10.2)	9.3 (8.2–11)	0.2	0.3
**MCV** (fL)	79 (73–86)	79 (73.1–86)	80 (72–87)	0.4	0.5
**Reticulocytes** (g/L)	214.0 (136–318)	217.5 (144.5–316)	201(132–319)	0.3	0.2
**LDH** (UI/L)	405 (305–484)	383 (305–489)	407 (299.5–477.5)	0.1	0.1
**Total bilirubin** (μmol/L)	35 (22–58)	35.5 (24.5–67)	33 (19–54)	0.5	0.6
**HbA2** (%)	3.4 (3–3.9)	3.4 (3–3.6)	3.5 (3–4)	0.7	0.4
**HbF** (%)	4.8 (2.3–8.2)	4.2 (1.7–8)	5.6 (2.9–8.2)	0.2	0.4

SCD: sickle cell disease, BMI: body mass index, Hb: hemoglobin, MCV: mean corpuscular volume, LDH: lactate dehydrogenase, and HbA2: adult hemoglobin A2, HbF: fetal hemoglobin. These biological features were determined prior hydroxyurea or transfusion program therapy at a steady state of the disease. Continuous variables are expressed as the median (IQR) or mean ± standard deviation. * multivariate analysis with adjustment for age and gender. ** multivariate analysis with adjustment for age, gender and genotype. Significant *p* values appear in bold.

**Table 2 jcm-08-02173-t002:** Comparison of acute complications of sickle cell disease according to the birth region.

Acute Complications	Whole Population *n* = 235	Patients Born in France *n* = 111	Patients Born in Sub-Saharan Africa *n* = 124	*p* ^a^	*p* ^b^
**Vaso-occlusive crisis**					
Number of admissions for VOC in the last 12 months	1 (0–2)	1 (1–2)	1 (0–2)	0.7	0.8
**Acute chest syndrome**, *n* (%)	162/231 (70.1)	86/110 (78.2)	76/121 (62.8)	**0.01**	**0.01**
Number of episodes over lifetime	1 (0–3)	2 (1–4)	1 (0–3)	**<0.001**	**0.002**
Age at first episode (years)	21 (16.4–26)	19 (11.6–22.3)	24 (18.4–29.5)	**<0.001**	**0.007**
≥ 1 ICU admission for ACS over lifetime	95/216 (44)	57/107 (53.3)	38/109 (34.9)	**0.006**	**0.006**
**Priapism**, *n* (%)	36/99 (36.4)	20/56 (35.7)	16/43 (37.2)	0.9	0.5
Age at 1st priapism episode (years)	16.8 (12.3–22.5)	17 (12.2–22.7)	15.4 (12.3–18.6)	0.8	0.8
**Stroke**, *n* (%)	19/224 (8.5)	6/107 (5.6)	13/117 (11.1)	0.1	0.2
Age at first stroke (years)	20.3 (11.1–33.5)	17.1 (12.6–21.5)	23.2 (11.1–33.9)	0.3	0.1
**Splenic complications**					
Splenic sequestration, *n* (%)	15/223 (6.7)	8/105 (7.6)	7/118 (5.9)	0.6	0.4
Splenectomy, *n* (%)	15/223 (6.7)	8/105 (7.6)	7/118 (5.9)	0.6	0.4
Age at splenectomy (years)	11.3 (9–14.8)	12.1 (6.9–16)	10.9 (9.5–13.5)	0.9	0.3
**Cholecystectomy**, *n* (%)	134/224 (59.6)	73/109 (67)	61/116 (52.6)	**0.03**	**0.04**
Age at cholecystectomy (years)	17 (12.6–23.8)	14.8 (10.9–20.8)	21.7 (15.5–30.5)	**<0.001**	0.5
**Thrombo-embolic events**, *n* (%)	20/230 (8.7)	9/107 (8.4)	11/123 (8.9)	0.9	0.1
Pulmonary embolism, *n* (%)	15/229 (6.6)	7/107 (6.5)	8/122 (6.6)	1	0.3
Age at 1st thrombo-embolic event (years)	28.2 (24.1–32.7)	29.2 (23.8–30.7)	26.9 (24.4–35.1)	0.7	0.2

VOC: vaso-occlusive crisis and ACS: acute chest syndrome. ICU: intensive care unit. Continuous variables are expressed as the median (IQR). ^a^ bivariate analysis: Chi^2^, Fisher’s, Student’s or Wilcoxon test as appropriate. ^b^ multivariate analysis with adjustment for age, gender and genotype. Significant *p* values appear in bold.

**Table 3 jcm-08-02173-t003:** Comparison of chronic complications of sickle cell disease according to country of birth.

Chronic Complications	Whole Population *n* = 235	Patients Born in France *n* = 111	Patients Born in Sub-Saharan Africa *n* = 124	*p* ^a^	*p* ^b^
**Retinopathy**, *n* (%)	103/207 (49.8)	51/101 (50.5)	52/106 (49.1)	0.8	0.4
Age at diagnosis of retinopathy (years)	24.7 (20.3–30.5)	21.2 (16.7–25.7)	28.9 (24.2–34.3)	**<0.001**	0.05
Laser photocoagulation, *n* (%)	52/185 (28.1)	25/90 (27.8)	27/95 (28.4)	0.9	0.1
**Cardiac involvement**, *n* (%)	67/177 (37.9)	36/85 (42.4)	31/92 (33.7)	0.3	0.6
Age at diagnosis of cardiopathy (years)	26.3 (20.6–30.4)	24.5 (20.5–28.7)	27.7 (21.2–31)	0.1	0.6
LV systolic dysfunction, *n* (%)	14/228 (6.1)	8/107 (7.5)	6/121 (5)	0.5	0.1
LV and/or LA dilatation, *n* (%)	56/161 (34.8)	31/82 (37.8)	25/79 (31.6)	0.4	0.7
Age at diagnosis of LV dilatation (years)	25.2 (20.4–30.1)	24.5 (18.7–28.7)	27.7 (21–30.8)	0.09	0.4
TRV ≥ 2,5 m/s, *n* (%)	33/86 (38.4)	14/41 (34.1)	19/45 (42.2)	0.4	0.9
**Cerebral vasculopathy**, *n* (%)	33/102 (32.3)	13/52 (25)	20/50 (40)	0.1	0.2
Brain aneurysms, *n* (%)	14/98 (14.3)	4/52 (7.7)	10/46 (21.7)	0.08	0.08
Silent cerebral infarcts, *n* (%)	10/85 (11.8)	6/45 (13.3)	4/40 (10)	0.6	0.4
Moyamoya, *n* (%)	2/97 (2.1)	1/54 (1.9)	1/43 (2.3)	0.9	0.9
Vessel stenosis, *n* (%)	7/98 (7.1)	2/52 (3.8)	5/46 (10.9)	0.2	0.3
**Nephropathy**, *n* (%)	64/164 (39)	31/84 (36.9)	33/80 (41.3)	0.6	0.4
eGFR (mL/mn/1.73 m^2^)	124 (109–134)	127 (116–136)	119 (99–131)	**0.02**	0.1
Hyperfiltration ^§^, *n* (%)	60/220 (27.3)	30/105 (28.6)	30/115 (26.1)	0.8	0.5
Chronic kidney insufficiency, *n* (%)	7/220 (3.2)	0/105 (0)	7/108 (6.1)	**0.01**	1
ACR > 3 mg/mmol, *n* (%)	53/144 (36.8)	27/73 (37)	26/71 (36.6)	0.9	0.9
ACR (mg/mmol)	1.8 (0.8–6.8)	1.5 (0.7–4.8)	2 (0.8–10)	0.5	0.4
**Bone complications**					
Avascular osteonecrosis (AON), *n* (%)	70/217 (32.3)	33/105 (31.4)	37/112 (33)	0.8	0.7
Age at diagnosis of AON (years)	19.8 (16.7–28.1)	18.9 (16.5–25.7)	21.2 (16.9–31.1)	0.06	0.4
H-shaped vertebrae, *n* (%)	47/88 (53.4)	20/42 (47.6)	27/46 (58.7)	0.3	0.2
Fracture, *n* (%)	57/181 (31.5)	32/85 (37.6)	25/96 (26)	0.09	0.1
Osteomyelitis, *n* (%)	45/214 (21)	25/96 (26)	20/118 (16.9)	0.1	0.06
Age at 1st osteomyelitis episode (years)	10.8 (6.6–19.4)	8.2 (2.8–19.9)	12 (8–18.3)	0.3	0.4
**Skin ulcers**, *n* (%)	24/188 (12.8)	6/95 (6.3)	18/93 (19.4)	**0.007**	**0.03**
Age at first episode (years)	18.6 (16.4–26.8)	23.2 (18.6–29.6)	18.5 (13.7–26.7)	0.3	**0.01**

LV: left ventricle, LA: left atrial, TRV: tricuspid regurgitation velocity, ACR: urine albumin-to-creatinine ratio, AON: avascular; osteonecrosis, and eGFR: estimated glomerular filtration rate. ^§^ Renal hyperfiltration was defined as an eGFR ≥130 mL/min/1.73; m^2^ for women and ≥140 mL/min/1.73 m^2^ for men. Chronic kidney insufficiency was defined by an eGFR < 60 mL/min/1.73; m^2^. Continuous variables are expressed as the median (IQR). ^a^ bivariate analysis: Chi^2^, Fisher’s, Student’s or Wilcoxon test. ^b^ multivariate analysis with adjustment for age, gender and genotype. Significant *p* values appear in bold.

**Table 4 jcm-08-02173-t004:** Comparison of treatments and of their side effects according to the region of birth.

Treatments	Whole population *n* = 235	Patients Born in France *n* = 111	Patients Born in Sub-Saharan Africa *n* = 124	*p* ^a^	*p* ^b^
**Hydroxyurea**					
Lifetime exposure, *n* (%)	135 (57.4)	70 (63.1)	65 (52.4)	0.1	0.2
Current treatment, *n* (%)	108 (46)	55 (49.5)	53 (42.7)	0.3	0.8
Age at introduction (years)	19.6 (15.4–25.3)	18.6 (14.4–23.5)	22.3 (16.4–30)	**0.0007**	**0.002**
Cumulative lifetime dose (g)	1299 (485–2453)	1299 (540–2748)	1306 (451–1954)	0.6	0.4
Cumulative lifetime duration (years)	4 (1.4–8)	3.8 (1.7–8.2)	4.1 (1.4–8)	0.7	**0.002**
**Blood transfusions**					
≥ 1 transfusion over lifetime, *n* (%)	190/216 (88)	94/104 (90.4)	96/112 (85.7)	0.3	0.9
≥ 1 transfusion in Africa, *n* (%)	20/54 (37)	1/13 (7.7)	19/41 (46.3)	**0.02**	**0.01**
**Chronic blood transfusion program**					
Previous, *n* (%)	63/222 (28.4)	29/106 (27.4)	34/116 (29.3%)	0.7	0.9
Current, *n* (%)	29/227 (12.8)	17/111 (15.3%)	12/116 (10.3%)	0.3	0.3
**Blood transfusion complications**, *n* (%)	44/195 (22.6)	24/96 (25)	20/99 (20.2)	0.4	0.3
Types of transfusion complications *				0.9	0.3
Antibodies without hemolytic reaction, *n* (%)	33/43 (76.7)	18/24 (75)	15/19 (78.9)		
DHTR with antibodies, *n* (%)	6/43 (14)	4/24 (16.7)	2/19 (10.5)		
DHTR without antibodies, *n* (%)	2/43 (4.7)	1/24 (4.2)	1/19 (5.3)		
Acute hemolytic transfusion reaction, *n* (%)	2/43 (4.7)	1/24 (4.2)	1/19 (5.3)		
**Proven hemochromatosis (liver biopsy or MRI)**, *n* (%)	28/212 (13.2)	13/104 (12.5)	15/108 (13.9)	0.8	0.4
Ferritin (μg/L)	109 (40–371)	99.0 (44–288)	116.5 (39.5–432.5)	0.4	0.3
Age at diagnosis of hemochromatosis (years)	18.8 (16.2–28.3)	16.9 (13.8–20.4)	25.3 (18.3–31.2)	**0.03**	**0.006**
**Morphine**					
Pruritus to intravenous morphine, *n* (%)	40/126 (31.7)	22/67 (32.8)	18/59 (30.5)	0.8	0.6
Chronic dependence on opioids, *n* (%)	22/150 (14.7)	14/79 (17.7)	8/71 (11.3)	0.3	0.1

DHTR: delayed hemolytic transfusion reactions and MRI: magnetic resonance imaging. Continuous variables are expressed as the median (IQR). ^a^ bivariate analysis: Chi^2^, Fisher’s, Student’s or Wilcoxon test. ^b^ multivariate analysis with adjustment for age, gender and genotype. * Precise type of complication was not specified for all patients in our database. Significant *p* values appear in bold.

**Table 5 jcm-08-02173-t005:** Comparison of viral infections in SCD patients according to their birth regions.

Infection Status	Whole Population *n* = 235	Patients Born in France *n* = 111	Patients Born in Sub-Saharan Africa *n* = 124	*p* ^a^	*p* ^b^
**Active or resolved HCV infection**, *n* (%)	16/174 (9.2)	3/84 (3.6)	13/90 (14.4)	**0.02**	0.1
**Active or resolved HBV infection**, *n* (%)	21/179 (11.7)	1/85 (1.2)	20/94 (21.3)	**<0.0001**	**0.004**
**Positive HIV serology**, *n* (%)	3/174 (1.7)	0/81 (0)	3/93 (3.2)	0.2	1
**Positive PVB19 serology**, *n* (%)	52/75 (69.3)	30/42 (71.4)	22/33 (66.7)	0.7	1
**Positive HTLV1 serology**, *n* (%)	1/68 (1.5)	0/31 (0)	1/37 (2.7)	1	1

HCV: hepatitis C virus, HBV: hepatitis B virus, HIV: human immunodeficiency virus, PVB19: parvo virus B19, and HTLV1: human T lymphotropic virus. ^a^ bivariate analysis: Chi^2^, Fisher’s, Student’s or Wilcoxon test. ^b^ multivariate analysis with adjustment for age, gender and genotype. Significant *p* values appear in bold.

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
