# Peer review of "Comparison between Adult Patients with Sickle Cell Disease of Sub-Saharan African Origin Born in Metropolitan France and in Sub-Saharan Africa"

_jcm, 2019, doi:10.3390/jcm8122173_

Round 1

Reviewer 1 Report

The manuscript describe a well designed study describing differences between individuals with sickle cell diseases born in Africa verses France.  It is well written and the conclusions are appropriate based on the data.

The main limitation is the retrospective nature of the study where there may be both selection bias because of differential survival of young children, opportunity for migration, and quality of the retrospective nature of the data from Africa.  These limitations are appropriately acknowledged, but reduce the generalizability and impact of the observations.

Author Response

The manuscript describe a well designed study describing differences between individuals with sickle cell diseases born in Africa verses France.  It is well written and the conclusions are appropriate based on the data.

Point 1 : The main limitation is the retrospective nature of the study where there may be both selection bias because of differential survival of young children, opportunity for migration, and quality of the retrospective nature of the data from Africa.  These limitations are appropriately acknowledged, but reduce the generalizability and impact of the observations.

Response 1 : Thank you for your comment. The only way to study the phenotype of SCD patients born in sub-Saharan Africa by avoiding this bias of differential survival in childhood would be to conduct prospective studies on the African continent.

Reviewer 2 Report

Thank you for the opportunity to review this revised manuscript. This revision is improved and the topic remains of interest, perhaps, but there are still some issues that must be addressed:

Lines 106 - 110 are very strange and unacceptable to this (American) reviewer. The authors assume that subjects born in the Carribbean would have admixture with Caucasian individuals...but not the subjects born in France? This is a curious assumption...For example, Thomas Jefferson brought Sally Hemmings to France in the 1700s. I am not familiar with a literature that suggests that having Caucasian ancestry modifies SCD severity nor that being Afro-French prevents having Caucasian ancestry. If patients born in the French Carribbean and etc are treated separately because of their country of origin, I suppose that's a little different since that does affect the study question which seems to be: "Does where you come from affect your SCD phenotype"?

2. The conclusions about acute chest syndrome and intensive care use seem to be confounded by length of follow-up. The years of follow-up within the center's care system would be needed to understand this data.

3. Suggest indicating significant differences in the tables using an asterisk.

4. It is interesting that there is no difference in lifetime hydroxyurea exposure between the groups. Suggests that SSA subjects are willing to take HU.

4. Suggest modifying the conclusion to be about something other than ACS given the confounders above. I suggest leaning on the title of the article and indicating that X, Y, Z complications were more common and A, B, C were less common.

Author Response

Thank you for the opportunity to review this revised manuscript. This revision is improved and the topic remains of interest, perhaps, but there are still some issues that must be addressed:

Point 1 : Lines 106 - 110 are very strange and unacceptable to this (American) reviewer. The authors assume that subjects born in the Carribbean would have admixture with Caucasian individuals...but not the subjects born in France? This is a curious assumption...For example, Thomas Jefferson brought Sally Hemmings to France in the 1700s. I am not familiar with a literature that suggests that having Caucasian ancestry modifies SCD severity nor that being Afro-French prevents having Caucasian ancestry. If patients born in the French Carribbean and etc are treated separately because of their country of origin, I suppose that's a little different since that does affect the study question which seems to be: "Does where you come from affect your SCD phenotype"? 

Response 1 : Thank you for this remark, which shows how unclear our sentence is. Our choice not to include patients from the West Indies does not come from the fact that they may have Caucasian ancestors. What we wanted to say is that the miscegenation of this population, by the history of the forced displacements of the African populations towards the Antilles in the 17th and 18th centuries, is very different from that of the African populations having migrated in France in the late 20th century. This element seems important to us in a disease where genetic determinants play a certain role (role which remains to be clarified). Following your comment, we modified the sentence in question, removing the reference part of the Caucasian ancestors : « Thus, we excluded patients born in the French West Indies, French Guiana and Reunion because we assumed that their genetic backgrounds were different from that of the SSA population due to admixture with Caucasian individuals.”

Point 3 : Suggest indicating significant differences in the tables using an asterisk.

Response 3 : Thank you for your suggestion. In order to highlight the significant results we chose to put them in bold rather than with an asterisk, the asterisk being already used in a previous table with a specific meaning.

Point 4 : It is interesting that there is no difference in lifetime hydroxyurea exposure between the groups. Suggests that SSA subjects are willing to take HU.

Response 4 : Indeed, this suggests that the later start of hydroxyurea could be due to its lack of accessibility in sub-Saharan Africa. The following sentence was added to the discussion: « The access to hydroxyurea is, until now, unafordable in SSA, due to its cost. Interestingly, our study shows no difference in lifetime hydroxyurea exposure between the groups, suggesting that SSA subjects are willing to take HU, when it is reimbursed by a care system”.

Point 5 : Suggest modifying the conclusion to be about something other than ACS given the confounders above. I suggest leaning on the title of the article and indicating that X, Y, Z complications were more common and A, B, C were less common.

Response 5 : We changed our conclusion as follows : "Compared to those of patients born in France, SCD patients born in SSA who emigrated in France unexpectedly had a less severe clinical phenotype in terms of ACS and intensive care admission for ACS regardless of their age, probably due to selection and survival bias. In contrast, skin ulcers occurred more frequently in patients born in sub-Saharan Africa."

Reviewer 3 Report

The authors of the paper did a complete and significant comparison of demographic, clinical and biological phenotypes of adult SCD patients of different place of birth. They have not forgotten the epidemiological aspect of SCD and the spread of HbS for migratory flows: these are aspects that cannot be considered about SCD.

The authors have clearly analyzed every aspect proposed in the study design and the limitations of their study.

I have some minor comments:

Lines 254-255: are the three transplanted patients born in France or in SSA? 

Line 296: please specify the genotype of the compound heterozygous patients, especially for “less severe phenotype”

Lines 340-342: please reformulate the sentence to be clearer

Table 3: 8/107 (7.3) (column of the patients born in France, line 7) 8/107= 7.47%=7.5%

Author Response

The authors of the paper did a complete and significant comparison of demographic, clinical and biological phenotypes of adult SCD patients of different place of birth. They have not forgotten the epidemiological aspect of SCD and the spread of HbS for migratory flows: these are aspects that cannot be considered about SCD.

The authors have clearly analyzed every aspect proposed in the study design and the limitations of their study.

I have some minor comments:

Point 1 : Lines 254-255: are the three transplanted patients born in France or in SSA? 

Response 1 : Of the four transplanted patients in our active file, three were born in sub-Saharan Africa, as stated at the end of the paragraph « Three of these 4 patients were born in SSA.”

Point 2 : Line 296: please specify the genotype of the compound heterozygous patients, especially for “less severe phenotype”

Response 2 : Thank you for this question, which shows how unclear our sentence is. By "less severe phenotype" we do not refer to compound heterozygotes, which represent only about 25% of our patients and whose mortality reported in the literature is very much lower than that of homozygotes. We were actually referring to other genetic determinants such as haplotypes that we can not detail further because only about 5% of our patients had their haplotype determined.

Point 3 : Lines 340-342: please reformulate the sentence to be clearer

Response 3 : The sentence has been reworded as follows : “In addition, it is possible that the SSA patients who were most likely to realize an emigration project and permanent settlement in France were more likely to receive education in their country of origin, compared to other patients having not emigrated.”

Point 4 : Table 3: 8/107 (7.3) (column of the patients born in France, line 7) 8/107= 7.47%=7.5%

Response 4 : Thank you for reporting this calculation error, which we have since corrected.

This manuscript is a resubmission of an earlier submission. The following is a list of the peer review reports and author responses from that submission.

Round 1

Reviewer 1 Report

This is an interesting study that compares adult subjects with SCD born in France to those born in sub-saharan Africa. This study provdies an important glipse into an emerging epidemiologic and clinical reality: a growing number of immgirants with SCD are coming from under-resourced to well resourced settings.

As it stands, this paper is significantly limited by two issues.

1. The paper does not attend to how sickle-cell type (compound heterozygous versus homzougous disease) contributes to the kinds of disease complications that patients experience.

For example: the authors conclude there is no difference in stroke between the two groups. Stroke is much more common in people with homozygous sickle cell disease. For the purposes of this thought experiment, I assumed that all the subjects with a history of stroke had homozygous disease (because this is most likely the case). When I assumed that all the subjects with stroke had HbSS or HbS-Beta-Zero, I saw that 6% (6/93) of French-born patients had a history of stroke and 14% (13/93) of the SSA-born patients had stroke. That's more than twice the number (did not run p-values). It is possible that this is a faulty assumption, but that would need to be shown. A similar logic might be applied to the problem, with very similar numerical outcomes, to the chronic kidney insufficiency which occurs more often in homozygous than compound heterozygous disease. [For more discussion of compound heterozygous disease vs homozygous disease see: https://doi.org/10.1111/bjh.14444 ]

In any case, to provide comparisons of disease complications really requires comparing apples to apples. Since the authors have chosen an important subject to study, I strongly encourage them to re-build their datasets to distinguish between compound heterozygotes and homozygote SCD. This would allow for a more clear comparison of phenotypic differences between groups. Since we do not offer the same interventions, do not see the same kinds of complications in compound heterozygous SCD and homozygous SCD, this is a really important distinction. There were more SSA-born subjects with compound heterozygous SCD than French-born subjects (31 vs 18). Since compound heterozygous disease is "less severe" and usually associated with a higher baseline hemoglobin, one wonders how this contributes to the conclusions...and if it doesn't, this should be addressed. 

2. The paper highlights an identified difference in ACS rates. This conclusion is confounded by (a) lack of transparency about the kinds of SCD in which these ACS events occurred and lack of engagement with the problem that hospitalizations of any sort are a risk factor for ACS. Until the issues related to other complicatiosn (like strong analyzed by disease type) are sorted out, the data do not support the conclusion that this is a phenotypic difference.

Two additional observations:

1. In countries without newborn screening, it is not uncommon for people with compound heterozygous disease to go undiagnosed for long periods of time. Therefore, one does wonder, are people with compound heterozygous disease from SSA arriving in France and not pursuing / receiving care due to lack of a diagnosis? Is it possible that the subjects you meet actually have a more severe phenotype compared to other (non-desceased) people with SCD born in SSA because they have had sufficient disease complications to present for care?

2. Differences in the development of leg ulcers is an interesting and might be further explored. There is a literature that suggests this finding is more common in under-resourced settings. Your data may be consistent with this...or have the subjects with leg ulcers born in SSA developed those ulcers in France? This would be an interesting variable to offer (leg ulcer in SSA vs France).

Overall, it is certainly intriguing to consider that subjects who survive childhood with SCD and are able to migrate to France as adults have a survival advantage over those who are unable to make this (often treacherous) journey. For this reason, I am not surprised that subjects with compound heterozygous disease are greater in the SSA-born group. It is also intriguing to wonder if those born in SSA therefore have a different phenotype than subjects who survive from early childhood because they have the good fortune to be born in France with access to world-class SCD care. However, conclusions such as these can only be drawn from more precise comparisons between subjects with the same kind of disease are made.

Reviewer 2 Report

This study characterizes the differences in two populations of subjects with similar geographic heritage; one born in France and one migrating from sub-Saharan Africa. It is a single-center observational study of 235 subjects.

The study extremely well presented and analysis complete.

The manuscript in well written and observations and conclusions well presented.

There are several limitations:

First, there appears to be selection bias because the ages and sex distribution of the two populations is very different. The population is highly selective both by geography and attendance at one medical Center. This makes generalizability very limited. The potential from selection by death in SSA and differential migration based on social resources makes it impossible to differentiate social and economic factors from biological differences. Certainly, the data may be of local interest, but likely not relevant to other population of patients.

Genetic admixture with other populations in the Paris cohort is not considered or discussed as a limitation.